# Enhancing Interpretation of Histopathology Whole Slide Image Analysis via Regional Causal Dependency Discovery

**Zixian Li[1]**                                    ZIXIANLI@BUAA.EDU.CN
**Jun Shi[2]**                                       JUNS@HFUT.EDU.CN
**Zhiguo Jiang[3]**                                 JIANGZG@BUAA.EDU.CN
**Fengying Xie[3]**                                  XFY_73@BUAA.EDU.CN
**Yushan Zheng[1],✉**                              YSZHENG@BUAA.EDU.CN

[1]*School of Engineering Medicine, Beihang University*
*Beijing, China*
[2]*School of Software, Hefei University of Technology*
*Hefei, China*
[3]*School of Astronautics, Beihang University*
*Beijing, China*

## Abstract

Histopathology whole slide image (WSI) analysis is fundamental to computational pathology. Attention-based heatmaps are commonly used for interpretability in WSI analysis. However, heatmap is limited in describing the potential relationships between multiple high-probability regions, which restricts its application in fine-grained WSI analysis tasks. In this paper, we propose Pathology Causal Discovery Network (PCDN), a novel framework that reconstructs interpretable diagnostic pathways by dynamically discovering regional causal dependencies from WSIs. Unlike approaches relying on predefined medical priors, PCDN introduces a Causal Structure Learner (CSL) to infer a Directed Acyclic Graph (DAG) which represents the causal dependencies among pathological regions. A Causal Graph Propagator (CGP) is then designed to guide feature propagation based on the DAG, integrating local causal dependencies with global context. Extensive experiments on three large-scale pathological datasets demonstrate that PCDN achieves state-of-the-art performance and can provide meaningful causal insights for WSI analysis.

**Keywords:** Computational pathology, WSI analysis, Causal Discovery

## 1 Introduction

Computational pathology has emerged as a promising technique, utilizing deep learning to automate whole slide image (WSI) analysis Song et al. (2023). One widely used approach is multiple instance learning (MIL) Shi et al. (2020); Gadermayr and Tschuchnig (2024); Qu et al. (2024); Lse et al. (2018); Lu et al. (2021); Shao et al. (2021), which treats WSI patches as instances within a bag and aggregates patch-level features. Although MIL holds significant potential for automating WSI analysis, it is limited in providing detailed and interpretable insights into what is the specific process to reach the prediction.

In computational pathology, achieving interpretability at the WSI level has been approached primarily through two main strategies. The first focuses on localizing decision-critical regions within WSIs. Techniques such as attention-guided visualizations Lse et al. (2018); Lu et al. (2021); Shao et al. (2021), and gradient-based saliency methods Zhou et al.

(2016); Selvaraju et al. (2017); Chattopadhay et al. (2018) can generate spatial heatmaps to highlight patches that significantly influence model predictions. Although these methods provide valuable transparency at the patch level, their clinical utility is limited by two factors: (1) the highlighted regions often do not directly correspond to recognizable histopathological entities, requiring pathologists to further interpret these regions with their domain knowledge; and (2) these techniques emphasize local morphological features, neglecting broader tissue-level context. The second approach employs a feature-based explanation framework, where predefined morphological features (e.g., nuclear circularity, glandular architecture, or cell density) are extracted from regions of interest and statistically related to clinical outcomes Kapse et al. (2024). Although this strategy follows traditional pathological reasoning, it is limited by the reliance on manually defined features.

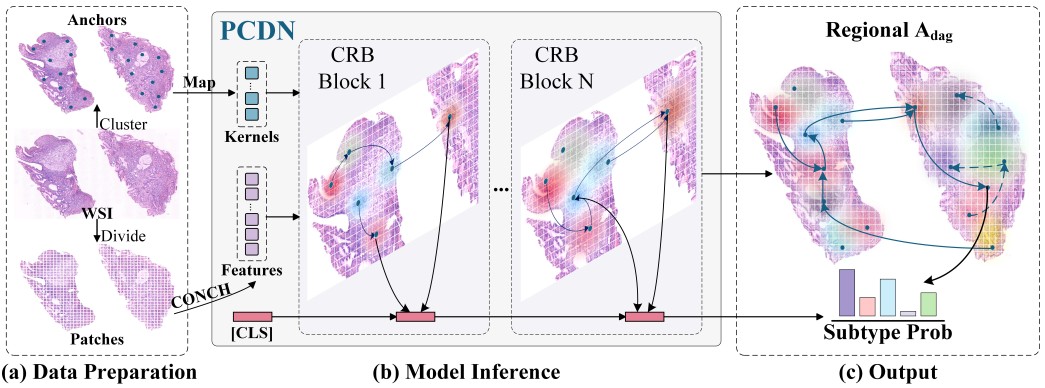

Figure 1: The proposed pathology causal discovery network (PCDN). (a) Extracted features and defined anchors serve as the input for the PCDN. (b) PCDN consists of $N$ causal representation blocks to discovery regional causal dependencies hierarchically. (c) The inferred causal graph of regions $\mathbf{A}_{dag}$ guides the prediction of the model.

In recent years, there has been growing interest in integrating causal inference methods into computational pathology models to enhance their interpretability Chan et al. (2023); Chen et al. (2024); Song (2024). One major challenge of existing methods is their reliance on predefined causal assumptions. In addition, they struggle to uncover and represent the causal dependencies among pathology regions, making it difficult to achieve transparent and explicit interpretability in the WSI analysis.

In this paper, we propose pathology causal discovery network (PCDN), a novel framework that reconstructs interpretable diagnostic pathways by discovering regional causal dependencies from WSIs, without relying on predefined causal assumptions. As shown in Fig. 1, PCDN maintains a Directed Acyclic Graph (DAG) that explicitly captures the causal dependencies among pathological regions throughout the WSI analysis process. Extensive experiments conducted on three large-scale pathology datasets demonstrate that PCDN achieves state-of-the-art (SOTA) performance while also offering causal insights with clinical relevance. The contributions of this paper can be summarized in two aspects.

1) We propose PCDN, a novel interpretable framework for WSI analysis. PCDN dynamically uncovers causal dependencies between pathological regions without relying on

predefined medical knowledge. This approach offers clearer and more intuitive insights for diagnostic reasoning compared to traditional interpretability methods and meanwhile improves the overall performance of WSI analysis.

2) A Causal Structure Learner (CSL) is designed to generate the DAG that represents spatial causal dependencies within the WSI. Meanwhile, a Causal Graph Propagator (CGP) is built to leverage the DAG to achieve the WSI reasoning. Importantly, the constructed DAG is not merely an explanatory tool but is directly integrated into the model's inference process, resulting in a unified approach that supports both prediction and interpretation.

## 2 Methods

### 2.1 Data Preparation

As shown in Fig. 1, patch-level features $\mathbf{X} \in \mathbb{R}^{n_p \times d}$ are extracted using a foundation model, where $n_p$ is the number of patches and $d$ is the feature dimension. To obtain region-level representations, we follow KAT Zheng et al. (2023) and apply K-means clustering to the spatial coordinates of foreground patches, generating $n_k$ anchors adaptively determined by each WSI's size. We then initialize $n_k$ learnable kernels $\mathbf{K} \in \mathbb{R}^{n_k \times d}$, each assigned to an anchor and trained to aggregate features within its spatial scope. The extent of each anchor's region is defined by a soft mask (see Appendix A),within which the associated kernel dynamically aggregates region-level features from surrounding patches.

### 2.2 Pathology Causal Discovery Network

As illustrated in Fig 2(a), the PCDN architecture operates hierarchically to model both local and causal dependencies. It takes as input the patch-level features $\mathbf{X}^0$ and the kernels $\mathbf{K}^0$. The first block in the network is a local representation block (LRB). LRB introduces a kernel attention (KA) module Zheng et al. (2023), where kernels interact with patch tokens through cross-attention to adaptively aggregate region-specific features for further causal analysis (see Appendix A). Subsequently, the model passes through $N$ causal representation blocks, each of which consists of an LRB, a CSL, and a CGP.

### 2.3 Causal Structure Learner (CSL)

The CSL module is responsible for discovering causal dependencies among pathology regions. We adopt a score-based causal discovery framework, which searches for DAGs that best explain the observed dependencies under assumptions such as causal sufficiency and the causal Markov condition Pearl (2000). To integrate causal discovery into the model, we impose a DAG constraint on the self-attention mechanism to filter spurious correlations and encourage structured directional dependencies. Initially, the current kernels and a special class token $[CLS]$ are concatenated to create the input $\mathbf{K}_{cls} \in \mathbb{R}^{(n_k+1) \times d}$. Then, the DAG is estimated directly from $\mathbf{K}_{cls}$ by self-attention mechanism:

$$\begin{aligned}
\mathbf{A}_{dag} &= StableAtt(\mathbf{Q}, \mathbf{K}), \\
\mathbf{Q} &= \mathbf{W}_q \mathbf{K}_{\text{cls}}, \quad \mathbf{K} = \mathbf{W}_k \mathbf{K}_{cls},
\end{aligned} \tag{1}$$

where $\mathbf{W}_q, \mathbf{W}_k \in \mathbb{R}^{d_e \times d_h}$ are weight matrices, $d_e$ is the embedding dimension, $d_h$ is the head dimension, and $StableAtt$ means stable attention operation Jin et al. (2024), which

is applied to obtain normalized attention scores between nodes. It is notable that the role

Figure 2: The structure of pathology causal discovery network (PCDN). Input features pass through the local representation block (LRB) and causal representation block CRB. A causal structure learner (CSL) and a causal graph propagator (CGP) work together to learn and propagate causal dependencies across the pathology regions.

of $\mathbf{A}_{dag}$ is to construct the causal dependencies among kernels and those between kernels and $[CLS]$. Therefore, we need to add a directed acyclic constraint to ensure it explicitly describes the causal dependencies Zheng et al. (2018), which is defined as

$$L_{\text{dag}} = \frac{\rho}{2}h_{\text{v}}^2 + \alpha h_{\text{v}}, h_{\text{v}} = \text{tr}(\exp(\mathbf{A}_{dag} \circ \mathbf{A}_{dag})) - (n_k + 1), \tag{2}$$

where $tr$ denotes the matrix trace, $\circ$ is the Hadamard product, and $\exp(\cdot)$ is the matrix exponential. $\rho$ and $\alpha$ are dynamic parameters that adjust based on the degree of $h_{\text{v}}$. If $h_{\text{v}}$ exceeds a predefined threshold, both $\rho$ and $\alpha$ are updated as $\rho \leftarrow \min(\rho \times 10, \rho_{\max})$, and $\alpha \leftarrow \alpha + \rho \times h_{\text{v}}$. $L_{\text{dag}}$ penalizes any violations of the expected causal dependencies among regions, guiding the self-attention mechanism to model these dependencies as edges in a DAG. Specifically, the vertical axis of the $\mathbf{A}_{dag}$ matrix represents the causal source nodes, the horizontal axis represents the target nodes, and a non-zero element at position $(i, j)$ indicates that node $i$ is a causal source of node $j$, with the value reflecting the strength of the causal dependencies.

Notably, the positions corresponding to $[CLS]$ as a causal source node in $\mathbf{A}_{dag}$ are masked to ensure that $[CLS]$ functions solely as a causal sink. It aggregates global causal dependencies without affecting the causal weights of the region nodes, thereby preventing any ambiguity in causal directionality.

## 2.4 Causal Graph Propagator (CGP)

The CGP facilitates the dynamic propagation of region-specific features across the learned causal graph. To ensure stable feature propagation, we normalize the DAG into a Laplacian matrix $\tilde{\mathbf{A}}_{dag}$ in accordance with graph convolutional (GCN) Kipf and Welling (2017) theory, which is then used with $\mathbf{K}_{cls}$ to propagate features through the causal graph. Propagation is achieved using several GCN layers defined as:

$$\mathbf{H}_{l+1} = \text{ReLU}(\tilde{\mathbf{A}}_{dag}\mathbf{H}_l\mathbf{W}_l), \quad l = 0, 1, ..., n_l, \tag{3}$$

where $\mathbf{H}_l$ represents the feature matrix at the $l$-th layer. Specifically, the input $\mathbf{H}_0 = \mathbf{K}_{cls}^{(i-1)} \in \mathbb{R}^{(n_k+1)\times d}$ represents the updated region representation within the class token $[CLS]^{(i)}$. Due to the masking strategy in CSL, $[CLS]^{(i)}$ does not participate directly in local node feature updates, instead, its representation reflects only the cumulative effect of causal contributions.

## 2.5 Loss Function

The total loss $L$ is defined as the sum of the cross-entropy loss $L_{\text{ce}}$ between the ground-truth $\mathbf{y}$ and the output logits $\mathbf{z}_c$ of the class token and the DAG loss $L_{\text{dag}}^{(i)}$ in each CSL block, weighted by a hyperparameter $\lambda$ to control the trade-off between the two components:

$$L = L_{\text{ce}} + \lambda \sum_{i=1}^{N} L_{\text{dag}}^{(i)}, \quad L_{\text{ce}} = -\mathbf{y}^T \log(\text{softmax}(\mathbf{z}_c)). \tag{4}$$

## 3 Experiments

### 3.1 Experimental Settings

The proposed method was evaluated on two public available datasets TCGA-RCC and TCGA-EGFR and an in-house dataset Gastric-2k. TCGA-RCC contains 940 cases with 3 subtypes of renal cell carcinoma. TCGA-EGFR contains 696 WSIs used for classifying EGFR mutations in lung adenocarcinoma, categorized into 4 classes: EGFR-19del, EGFR-L858R, Wild, and Other types. Gastric-2K is an in-house dataset that consists of 2040 WSIs from 6 categories of gastric pathology: low-grade intraepithelial neoplasia (LGIN), high-grade intraepithelial neoplasia (HGIN), adenocarcinoma (A.), mucinous adenocarcinoma (MA), signet-ring cell carcinoma (SRCC), and non-tumor tissue (Normal).

Each dataset was randomly split into training and test sets in a 7:3 ratio. Five-fold cross-validation was conducted within the training set, with early stopping and hyperparameter selection based on validation data in each fold. Final evaluation was performed on the test set. Baseline methods were configured according to their original implementations.

Patch features were extracted under 20x lenses with CONCH Lu et al. (2024), where patch size is 256 and feature dimension is 512. The number of CRBs was empirically set to 3. The DAG loss weigh $\lambda$ is robust within the range of $[1, 10]$ and set to 5. The model was trained with a learning rate of $1 \times 10^{-4}$ using the Adam optimizer. The initial values of $\rho$ and $\alpha$ were 1.0 and 0.0 respectively, with $\rho_{\max} = 10^6$. All experiments were conducted in Python using PyTorch and run on a computer with Nvidia GeForce 4090 GPUs.

Table 1: Results of hyperparameter settings and ablation study ($n_l$: number of GCN layers). Best values are in bold and second-best values are underlined.

| Settings | AUC (%) | ACC (%) | F1 (%) | FLOPs ($\times 10^9$) | Mem. (MB) |
|---|---|---|---|---|---|
| PCDN ($n_l = 1$) | 93.80 (2.28) | 90.16 (1.80) | 71.16 (2.00) | **27.455** | **108.36** |
| PCDN ($n_l = 2$) | 94.64 (1.00) | 90.50 (1.60) | 70.00 (6.30) | 27.460 | **108.36** |
| PCDN ($n_l = 3$) | **95.18** (1.30) | 90.36 (1.70) | **75.78** (4.20) | 27.473 | 108.62 |
| PCDN ($n_l = 4$) | 93.28 (1.90) | **90.94** (1.70) | 72.38 (6.50) | 27.486 | 108.89 |
| PCDN ($n_l = 5$) | 95.12 (1.50) | 90.22 (2.00) | 72.00 (8.10) | 27.499 | 109.15 |
| PCDN ($n_l = 3$) w/o $L_{\text{dag}}$ | 93.88 (1.00) | 90.12 (1.60) | 69.20 (7.70) | 27.473 | 108.62 |
| PCDN ($n_l = 3$) w/o CGP | 91.84 (3.40) | 89.00 (2.00) | 67.74 (8.40) | 27.477 | 109.28 |

## 3.2 Model Structure Verification

**Hyperparameter settings.** We first adjust the number of GCN layers in the CGP module. The cross-validation results are presented in Table 1, which shows that the best performance was achieved with three GCN layers ($n_l = 3$). These results suggest that three iterations of causal inference is the most effective for summarizing the regional information of the WSIs. FLOPs and memory usage increase slightly with more layers (FLOPs: $27.455 - 27.499 \times 10^9$; memory: $108.36 - 109.15MB$), but the overhead remains minimal. Based on the results, we fixed $n_l = 3$ in the following experiment.

**Ablation study.** $L_{\text{dag}}$ constraint Table 1 enforces the self-attention mechanism to uncover the causal dependencies among kernels. When removed, the model focuses only on kernel correlation. This leads to a 1.3% decrease in AUC and a 6.58% drop in F1-score, indicating that its removal introduces redundancy and noise. In the w/o CGP experiment, regional features are propagated via simple matrix multiplication based on $\mathbf{A}_{dag}$, leading to a 3.34% decrease in AUC, an 8.04% drop in F1-score and increased FLOPs and memory usage. Moreover, CGP's multi-layer propagation supports cross-layer causal interactions, and its removal eliminates hierarchical reasoning. Consequently, $L_{\text{dag}}$ ensures the causal structure's validity and sparsity, while CGP enables directed regional features propagation. They jointly ensure that the model extracts high-confidence causal dependencies from WSIs.

**Visualization with local causal dependencies.** Fig. 3 provides an application example of a gastric case with the trained PCDN. To highlight the most important causal dependencies among the nodes (i.e., region-level feature representations for each anchor), low-weight edges in $\mathbf{A}_{dag}$ are removed by thresholding. In Fig. 3d and e, the inferred DAG describes third-order causality. First, Anchors *1*, *3* and *6* are identified as initial causal sources, corresponding to dysplastic areas indicative of a precancerous state, and directed towards Anchor *9*. Anchor *9* exhibits faint but preserved glandular structures, suggesting well-differentiated adenocarcinoma. It further influences Anchor *2*, where fused cribriform glands indicate moderate differentiation. Finally, Anchors *9* and *2* serve as causal sources for the class token, where diagnostic information is summarized, and the WSI is classified as adenocarcinoma. The learned causal structure aligned with expert pathological assessment.

Compared to the conventional attention-based heatmap in Fig. 3c, which merely highlights discriminative areas, PCDN enhances interpretability by performing causal discovery

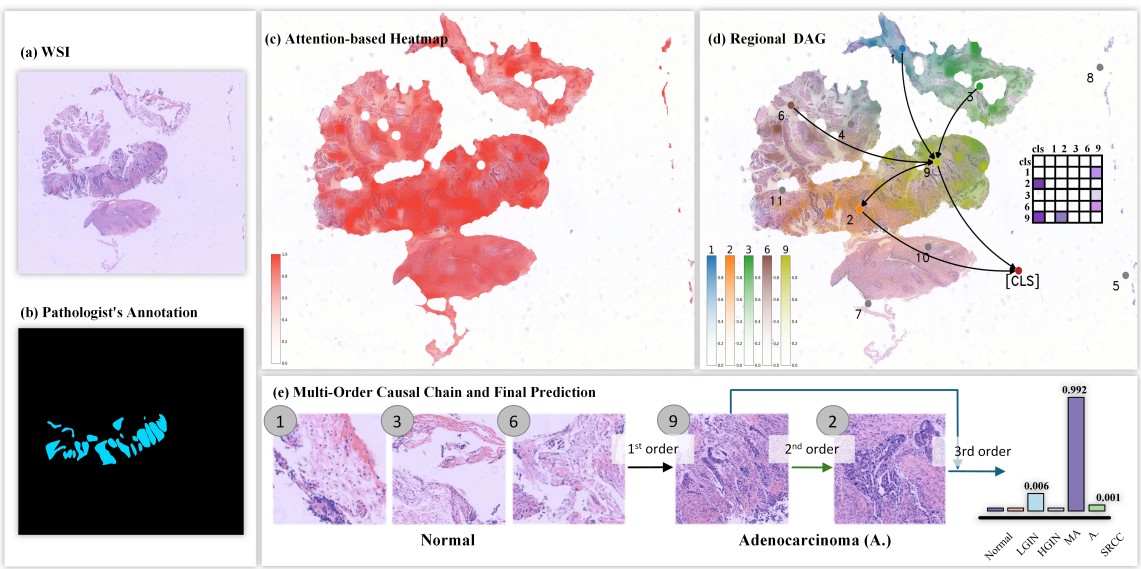

Figure 3: Visualization of causal graph on WSI based on PCDN, where (a) shows a raw WSI within adenocarcinoma, (b) presents tumor annotations by pathologists, (c) visualizes a conventional attention-based heatmap, (d) highlights the crucial regional representation and the discovered DAG with low-weight edges removed by thresholding. Distinct areas are color-coded and connected via inferred causal dependencies. (e) shows a multi-order causal chain and the final prediction.

to uncover meaningful dependencies among regions. The learned DAG helps filter spurious correlations and encourages structured, directional reasoning aligned with pathological progression. It also provides topological insights into WSI analysis, offering greater potential to support pathologists with more informative, actionable diagnosis, and research guidance.

### 3.3 Comparison with State-of-the-art methods

Finally, we evaluate PCDN against seven SOTA methods, covering various types: (1) Attention-enhanced frameworks: CLAM Lu et al. (2021), DTFD Zhang et al. (2022); (2) Context-aware methods: TransMIL Shao et al. (2021)(self-attention), MambaMIL Yang et al. (2024) (Mamba), and WiKG Li et al. (2024) (GNN); (3) Morphological prototyping method: PANTHER Song et al. (2024), and (4) a causal inference-based MIL model: IBMIL Lin et al. (2023). Table 2 presents the performance comparison on TCGA-EGFR, TCGA-RCC, and Gastric-2K datasets.

Overall, the proposed PCDN achieves performance comparable to SOTA methods across three datasets with diverse WSI analysis tasks. In the TCGA-RCC dataset, which presents a relatively simple task, all methods achieved satisfactory performance, owing to the discriminative patch features powered by CONCH. PCDN stands out by focusing on regions with causal dependencies, while disregarding causal chains with minimal influence on decision-

Table 2: Performance comparison with other methods on different datasets, where the best values are printed in bold, and the second best values are underlined.

| Method | TCGA-EGFR | | | Gastric-2K | | | TCGA-RCC | | |
|---|---|---|---|---|---|---|---|---|---|
| | AUC(%) | ACC(%) | F1(%) | AUC(%) | ACC(%) | F1(%) | AUC(%) | ACC(%) | F1(%) |
| CLAM Lu et al. (2021) | 73.04(2.54) | 65.22(4.50) | 69.24(3.77) | 87.52(1.36) | 87.78(1.25) | 61.56(4.74) | 96.06(0.97) | 90.08(1.59) | 87.52(2.07) |
| TransMIL Shao et al. (2021) | 51.36(5.85) | 75.64(2.40) | 69.80(1.32) | 83.34(3.82) | 85.44(1.63) | 48.40(5.61) | 98.26(0.72) | 91.26(0.54) | 89.60(1.01) |
| DTFD Zhang et al. (2022) | 72.66(1.93) | 77.20(1.29) | 76.36(0.77) | 87.48(2.52) | 87.78(1.10) | 64.40(3.45) | 96.96(0.19) | 92.74(0.63) | 90.64(1.39) |
| IBMIL Lin et al. (2023) | 72.58(3.88) | 78.94(1.31) | 74.28(1.92) | 92.10(1.01) | 88.66(0.62) | 58.20(4.27) | 98.58(0.29) | 91.58(0.70) | 89.58(0.85) |
| WiKG Li et al. (2024) | 68.54(6.24) | 75.08(2.76) | 73.06(2.34) | 91.38(0.30) | 88.96(1.21) | 63.80(6.20) | 98.42(0.70) | 91.52(1.32) | 89.88(1.66) |
| MambaMIL Yang et al. (2024) | 64.94(4.67) | 76.70(2.56) | 72.74(1.82) | 90.20(0.87) | 87.82(0.54) | 58.38(5.05) | 98.98(0.26) | 92.66(0.54) | 90.98(1.06) |
| Panther Song et al. (2024) | 66.72(4.66) | 78.74(1.72) | 73.66(1.94) | 91.82(0.51) | 89.78(0.65) | 57.14(1.74) | 98.90(0.19) | 91.00(0.81) | 91.00(0.01) |
| KAT Zheng et al. (2023) | 65.48(4.91) | 78.36(0.98) | 73.84(1.19) | 91.64(2.21) | 89.20(0.77) | 60.44(4.63) | 98.58(0.36) | 91.74(0.99) | 90.14(1.26) |
| PCDN | 74.06(2.01) | 80.58(0.94) | 77.28(1.58) | 93.30(0.94) | 89.02(0.75) | 65.24(0.89) | 98.80(0.24) | 92.82(0.71) | 91.32(1.08) |

making. Gastric-2K dataset which involves a more fine-grained 5-classes classification task, further highlights the strength of PCDN. By inferring regional causal dependencies, PCDN delivers the best AUC of 93.3% and F1-score of 65.24%, demonstrating its ability to capture complex interactions in multi-class scenarios. Finally the TCGA-EGFR dataset presents the most challenging gene-related task. PCDN excels in this task by modeling latent relationships between genes, yielding more accurate classification results with an AUC of 74.06%, an ACC of 80.58%, and an F1-score of 77.28%. IBMIL which incorporates interventional training to reduce confounding, demonstrates competitive performance. WiKG is a method based on undirected graph structure, which struggles to capture causal directionality, reducing its effectiveness. In contrast, PCDN enables end-to-end modeling of causal dependencies among pathological regions through dynamic causal discovery and directed propagation mechanisms. This approach allows PCDN to surpass traditional methods that primarily focus on simple correlations between local and global features. Notably, the proposed method can directly provide the causal graph that informs the decision-making process, offering a valuable tool to supplement heatmaps in interpretability.

## 4 Conclusion

We proposed the Pathology Causal Discovery Network (PCDN), a novel approach that improves whole-slide image (WSI) analysis by dynamically discovering causal dependencies between pathological regions. A causal structure learner (CSL) and a causal graph propagator (CGP) are then designed to infer a directed acyclic graph (DAG) for feature propagation. PCDN effectively constructs the underlying region dependencies in WSI through the designed causal structure learning approach. Experiments on three large-scale datasets demonstrate the effectiveness of PCDN and its potential to enhance the interpretability of WSI analysis. Future work will improve PCDN's causal modeling by moving beyond DAG-constrained attention, leveraging identifiable and interventional graph learning.

## Acknowledgments and Disclosure of Funding

This work was supported by the Beijing Natural Science Foundation (Grant No. 7242270), the Anhui Provincial Natural Science Foundation (Grant No. 2408085MF162), and the National Natural Science Foundation of China (Grant No. 62171007).

## Appendix A.

**Kernel Attention Module**. To capture region-level representations, the kernel attention (KA) module establishes bidirectional cross-attention between kernels and patches. This interaction is guided by a spatial soft mask $\mathbf{M} \in \mathbb{R}^{K \times n_p}$, where each entry $m_{ki}$ encodes the spatial weight between the $k$-th kernel and the $i$-th patch, computed based on their Euclidean distance:

$$m_{ki} = \exp\left(-\frac{\|p(\mathbf{f}_i) - \mathbf{c}_k\|_2^2}{2\delta^2}\right), \tag{5}$$

where $p(\mathbf{f}_i)$ and $\mathbf{c}_k$ denote the spatial coordinates of the $i$-th patch and the $k$-th kernel, respectively. The scaling parameter $\delta$ controls the spatial range of the Gaussian-like mask.

Firstly, each kernel receives information from its associated patches via cross-attention, denoted as:

$$\mathbf{K}^{(n+1)} = \sigma\left(\frac{\mathbf{K}^{(n)}\mathbf{W}_q^{(n)} \cdot \left(\mathbf{X}^{(n)}\mathbf{W}_k^{(n)}\right)^\top}{\sqrt{d_e}}\right) \odot \mathbf{M} \cdot \mathbf{X}^{(n)} \cdot \mathbf{W}_v^{(n)}, \tag{6}$$

where $\mathbf{X}^{(n)} \in \mathbb{R}^{n_p \times d}$ and $\mathbf{K}^{(n)} \in \mathbb{R}^{n_k \times d}$ denote the patch and kernel features at the $n$-th block. $\mathbf{W}_q^{(n)}, \mathbf{W}_k^{(n)}, \mathbf{W}_v^{(n)} \in \mathbb{R}^{d \times d_e}$ are learnable projection matrices for queries, keys, and values, with $d_e$ being the output dimension of each head, and $\sigma$ denotes the softmax function. Subsequently, each patch aggregates information from kernels to update its own local representation, formulated as:

$$\mathbf{X}^{(n+1)} = \sigma\left(\frac{\mathbf{X}^{(n)}\mathbf{W}_q^{(n)} \cdot \left(\mathbf{K}^{(n)}\mathbf{W}_k^{(n)}\right)^\top}{\sqrt{d_e}}\right) \odot \mathbf{M} \cdot \mathbf{K}^{(n)} \cdot \mathbf{W}_v^{(n)}. \tag{7}$$

This bidirectional interaction enables kernels and patches to exchange hierarchical spatial information, benefiting fine-grained region-level representation learning.

## Appendix B.

**Computational complexity**. Our DAG-based causal structure learning is built on Anchor Representations. Let $n_k$ denotes the anchor number and $n_p$ be the patch number, its complexity is $O(n_k^2)$, which is much more efficient than the $O(n_k \cdot n_p)$ of KAT block since $n_k \ll n_p$ (tens vs. thousands). As a result, the inference of PCDN requires 27.47 GFLOPs per WSI, which is only 0.51% higher than the 27.33 GFLOPs required by the KAT baseline.

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
