# OpenReview forum: "Enhancing Interpretation of Histopathology Whole Slide Image Analysis via Regional Causal Dependency Discovery"
_MICCAI.org/2025/Workshop/COMPAYL — COMPAYL 2025_

### Official Review · Reviewer_bWqq · 2025-07-11
**While the idea of dynamically inferring regional causal dependencies is novel and promising, several methodological and empirical concerns limit confidence in the claims.(Weakly Accept)**

**Rating:** 4
**Confidence:** 5

**Review:**

The paper proposes an innovative framework (PCDN) for interpretable WSI analysis via causal discovery, with solid experiments across three datasets. While the idea of dynamically inferring regional causal dependencies is novel and promising, several methodological and empirical concerns limit confidence in the claims.

Strengths
1. Novelty: The integration of causal discovery (CSL) with graph propagation (CGP) for WSI analysis is a meaningful advance over attention-based or correlation-driven methods.
2. Interpretability: The DAG-based explanations (e.g., Figure 3) offer a potential step toward actionable pathological insights, surpassing heatmap-only approaches.
3. Experiments: Evaluation on diverse datasets (TCGA-RCC, EGFR, Gastric-2K) demonstrates broad applicability, with competitive SOTA results in some tasks.

Limitations(Concerns)
1. How does the CSL module ensure that the inferred causal dependencies (DAG edges) reflect biologically or clinically meaningful relationships, rather than spurious correlations? Are the results validated by pathologists or grounded in established medical knowledge? The paper mentions avoiding predefined medical priors, but how does the model address potential confounders (e.g., tissue artifacts or batch effects) that could distort causal inference?
2. While the complexity is claimed as O(nk2)O(nk2​), how does the method scale to WSIs with extreme sizes (e.g., >100k patches) or higher-resolution scans? Are there runtime comparisons for large-scale deployments? The memory footprint (~109MB) seems reasonable, but does this account for GPU memory overhead during training with multiple concurrent slides?
3. In Table 2, PCDN achieves a notably higher F1-score (77.28%) on TCGA-EGFR (gene mutation classification). Since WSIs lack explicit molecular data, how does the model link morphological features to EGFR mutations causally? Are there visualizations or case studies to support this? Some baselines (e.g., TransMIL) perform poorly on TCGA-EGFR (AUC 51.36%). Were dataset splits or task settings consistent across methods? Could class imbalance or label noise explain these disparities?

---

### Official Review · Reviewer_uo8j · 2025-07-12
**In this study, authors present a novel framework for whole slide image (WSI) classification through causal dependency discovery. The proposed method dynamically identifies dependencies within WSIs using learnable anchors, aiming to enhance the interpretability of attention maps. The approach is evaluated on three datasets: TCGA-RCC, TCGA-EGFR, and Gastric-2K (an in-house dataset). While the framework is innovative, further clarification is needed on how proposed model is learning the relationship between the ordering of different kernels as presented (in figure 3) and how each kernel is back tracked to a specific patch for transparent interpretability in WSI analysis**

**Rating:** 5
**Confidence:** 5

**Review:**

This study introduces a novel framework for whole slide image (WSI) classification by utilizing causal dependency discovery with learnable anchor vectors. In this method, authors start with performing spatial clustering within WSI using K-means and associate each learnable kernel with each anchor (closest patch to each cluster centroid). Then, a cross-attention mechanism is used for each kernel to learn the representation for features of patches from its spatial cluster, and then dynamically models relationships within learnable kernels, aiming to establish a relationship between different regions e.g., (dysplastic areas -> well-differentiated Adenocarcinoma -> fused cribriform glands) and enhance the interpretability of attention maps, a significant advancement in transparent computational pathology. The methodology is rigorously evaluated across three diverse datasets (TCGA-RCC (sub-typing), TCGA-EGFR (multi-class mutation prediction), and the in-house Gastric-2K (six-class gastric pathology prediction), demonstrating robust classification performance.

# Weaknesses

1. Ambiguous Terminology & Methodology Gaps:
There are some ambiguities in the manuscript such as the claim that "each kernel is explicitly bound with an anchor" conflicts with Equation 6, which suggests kernels associate with patches within clusters, not anchors alone. Also, Additional clarification will be beneficial for the readers on how the network handles cases where anchors represent similar morphology across clusters as spatial clustering doesn’t take features/tissue morphology into account.

2. Unresolved Interpretability Mechanisms:
The manuscript lacks details on (1) how initial causal sources are identified, (2) relationship between each anchor and kernel as eq.6 uses multiple patches during cross-attention operation. Crucially, it does not explain based on kernels DAG matrix, how identified relationships are backtracked to specific patches for interpretability. The threshold for removing low-weight edges in A_dag is also not provided, and generalizability of threshold across slides.

3. Undefined Parameters & Experimental Shortcomings:
Key implementation details are missing such as: the value of K (anchor count) and its impact on performance, parameters (hv, ρ, α, p_max), and the threshold min(p × 10, p_max).

4. Presentation & Scope Issues:
References contain incomplete placeholders in references (e.g., "computational pathology. Unknown Journal, 2023"). The text incorrectly cites "two publicly available datasets" (should be three). A dedicated limitations section is absent.

Please find my detailed comments below:

## Major comments:

1.	The manuscript uses the terms “nodes” and “anchors” interchangeably. For clarity, please explain the difference between them or use a consistent terminology.
2.	Proposed methodology involves spatial clustering to select anchors, followed by dynamic modeling of relationships between them. Assume anchors correspond to same tissue morphology (because features are not seen); how will this network tackle this especially in a case where anchors represent a similar morphology among different clusters. Additional explanation would enhance understanding.
3.	Please confirm whether the sentence “we define a set of learnable kernels K ∈ Rⁿᵏ×ᵈ where each kernel is explicitly bound with an anchor” is correct. Based on the manuscript, it seems each kernel is associated with patches within a cluster, not with the anchor only (eq. 6).
4.	Since the classification (CLS) function is masked out, could the authors explain the rationale for using nk + 1 rather than nk? Please explain how the CLS function captures the cumulative effect of causal contributions in CGP, particularly how the CLS function is utilized in subsequent steps and its relevance to the final output.
5.	The manuscript does not clearly outline how the interpretability aspect is addressed. Please elaborate on how initial causal sources are identified, how relationships are established, and where anchor patches are linked with kernels.
6.	Study lacks the relationship between kernel and patch during interpretability. This is because kernel is learned due to its cross-attention within each cluster. Please explain how this process is back tracked and reached back to the patch level from the kernel’s regional DAG matrix.
7.	Please explain how the threshold for removing low-weight edges in A_dag was computed. Does this threshold generalize across different slides?
8.	Please indicate the value of K used in the experiments and how it impacts the performance of model.
9.	Can the authors comment on the applicability of this method to “needle-in-a-haystack” scenarios, such as detecting micro metastases in WSIs?
10.	The manuscript would benefit from a section discussing its limitations. Please consider adding a paragraph outlining potential constraints or areas for future improvement.
11.	The convergence of the method depends on parameters such as hv, ρ, α, and p_max. Please define how the threshold min (p × 10, p_max) is determined and their values used in the experimental settings.
12.	In the appendix section, are query, key, and value layers shared between (6) and (7)? Please define it in the manuscript.

## Minor comments:

1.	Please update the references containing incomplete or placeholder entries, such as: “computational pathology. Unknown Journal, 2023.”
2.	Please mention that additional details related to KA (2.2) are available in the appendix.
3.	Please correct the phrase “two publicly available datasets” to “three publicly available datasets.”
4.	In Table 1, please define what nl stands for in the caption.
5.	In appendix, it is mentioned that p(f_i) and c_k denotes spatial coordinates of i-th patch and k-th kernel. Aren’t Kernels learnable here?

---

### Official Review · Reviewer_YMrA · 2025-07-13
**Enhancing Interpretation of Histopathology Whole Slide Image Analysis via Regional Causal Dependency Discovery**

**Rating:** 2
**Confidence:** 4

**Review:**

Short summary: This paper introduces PCDN, a deep learning architecture for whole-slide image classification that integrates a learned DAG to model regional causal dependencies. The proposed approach combines a Causal Structure Learner and a Causal Graph Propagator to build a graph-based representation of inter-region relationships, aimed at improving both performance and interpretability. Results on three datasets show strong classification accuracy and F1 scores, and a visual example is provided to illustrate potential interpretability.

Strengths:
•	Technically sophisticated and original model architecture, with structured graph-based reasoning built into the inference process.
•	Novel attempt to incorporate causal structure learning for regional interpretability in WSI analysis.
•	Extensive experiments, including ablation studies, comparison with state-of-the-art methods, and visual illustrations.

Weaknesses:
•	The central claim of improved interpretability through causal discovery is not properly validated. No experiments test whether the learned DAGs reflect clinically meaningful reasoning or whether they help pathologists understand model decisions.
•	There is no interventional evaluation, user study, or agreement with expert annotations to support the idea that the DAG captures valid causal dependencies.
•	It remains unclear whether the DAG reflects causality in any meaningful sense, or whether it is just a regularized correlation structure.
•	The work appears to be an incremental extension of the authors’ prior kernel-based architectures and the added value of the DAG over previous models is not clearly demonstrated.
•	Given the large number of existing MIL-based WSI classifiers, the paper’s contribution hinges on the interpretability aspect, which remains underdeveloped.

The paper is technically well executed and builds on solid prior work, but the primary contribution, interpretable causal reasoning via DAG, is not convincingly demonstrated